# Yes, no, maybe? Revisiting language models' response stability under paraphrasing for the assessment of political leaning

**Patrick Haller, Jannis Vamvas, Lena A. Jäger**
Department of Computational Linguistics
University of Zurich
{haller, vamvas, jaeger}@cl.uzh.ch

## Abstract

An increasing number of studies are aimed at uncovering characteristics such as personality traits or political leanings of language models (LMs), using questionnaires developed for human respondents. From this previous body of work, it is evident that models are highly sensitive to prompt design, including the phrasing of questions and statements, as well as the format of the expected response (e.g., forced choice, vs open-ended). These sensitivities then often lead to inconsistent responses. However, most studies assess response stability on a small scale with low statistical power e.g., using less than ten paraphrases of the same question.

In this work, we investigate the stability of responses to binary forced-choice questions using a large number of paraphrases. Specifically, we probe both masked language models (MLMs) and left-to-right generative language models (GLMs) on the political compass test, assessing response validity (i.e., the proportion of valid responses to a prompt) and response stability (i.e., the variability under paraphrasing) across 500 paraphrases of each statement. This large-scale assessment allows us to approximate the underlying distribution of model responses more precisely, both in terms of the overall stability of a model under paraphrasing as well as the stability of specific items (i.e., the intended meaning of a question). In addition, to investigate whether there are structural biases that drive model responses into a certain direction, we test the association between different word- and sentence-level features, and the models' responses.

We find that while all MLMs exhibit a high degree of response validity, GLMs do not consistently produce valid responses when assessed via forced choice. In terms of response stability, we show that even models that exhibit high overall stability scores flip their responses given certain paraphrases. Crucially, even within-model, response stability can vary considerably between items. We also find that models tend to agree more with statements that show high positive sentiment scores.

Based on our results, we argue that human-centered questionnaires might not be appropriate in the context of probing LMs as both their response validity and stability differ considerably between items. Moreover, although stability metrics represent useful descriptions of model properties, it should be emphasized that even for models exhibiting fairly high stability, specific paraphrases can lead to substantially different model responses.[1]

## 1 Introduction

Recently, there has been a surge of work aimed at revealing properties of (large) language models (LMs) such as political leanings (Feng et al., 2023), personality traits, or other human characteristics (Jin et al., 2023; Scherrer et al., 2023), utilizing psychological instruments such as psychometric tests and questionnaires, developed to be conducted with human

---

[1]Code and data are available at https://github.com/ZurichNLP/llm-response-stability.

respondents. Feng et al. (2023) have shown that LMs trained on politically biased corpora can propagate social biases into downstream tasks, such as hate speech and misinformation detection tasks, and may significantly affect predictions.

A common method to apply psychometric instruments such as questionnaires to LMs is to prompt them with the items (i.e., questions, tasks, etc.) these instruments provide. In most cases, the LM is prompted to generate a response that represents the selected option. It has not gone unnoticed that the question-response format as well as the phrasing of questions and statements themselves can heavily affect model responses, and several studies have assessed response stability[2] under different categories of prompt perturbations, for instance, when varying the statement to respond to, or varying the instruction surrounding the statement (Röttger et al., 2024; Ceron et al., 2024; Shu et al., 2024; Feng et al., 2023, cf. §2.3). However, these assessments have been conducted on relatively small scales, quantifying response stability based on few conditions (e.g., less than 10 paraphrase of the same statement), and only on the group-, rather than the item-level. Moreover, their stability is analyzed after the mapping from continuous probabilities to discrete responses ("agree", "disagree", etc.), using measures such as Fleiss' $\kappa$ (Fleiss, 1971). Lastly, despite the knowledge of low response stability of some LMs, measures of uncertainty are often provided as informative sanity checks, but do not get actually factored in for the assessment of political leanings, for instance (Feng et al., 2023).

In this present work, we prompt LMs with binary forced-choice questions and investigate response validity and response stability under paraphrasing on a large scale, using items from the Political Compass Test [3]. Specifically, we ask the following research questions (RQ):

RQ 1: How large is the overall model and per-item response validity (i.e., whether the model returned a valid response) under forced choice?

RQ 2: How large is the response stability under paraphrasing of a given model (a) on average; i.e., how much do model responses vary across all paraphrases of all items, and (b) on the item-level; i.e., how much do the model responses vary for different paraphrases of a given item?

RQ 3: Are there specific items that show consistently low stability under paraphrasing across all models?

RQ 4: Are structural word- and sentence-level features such as sentence length, word length, lexical frequency, syntactic dependencies and sentiment associated with certain model responses?

To answer these questions, we automatically create 500 paraphrases of each questionnaire item[4] and extract model responses from both masked language models (MLMs) and left-to-right generative language models (GLMs). We find that although many models show relatively high response stability in terms of standard deviations, many items show low response stability. Moreover, we find a strong association between sentiment and model responses, suggesting that there are structural biases that guide the model towards specific responses.

In summary, we make the following contributions:

- We conduct the first large-scale assessment of response stability under paraphrasing (500 paraphrases), allowing us to quantify stability with high statistical power.
- An analysis of 18 LMs shows that most models exhibit rather poor stability under paraphrasing, both in terms of overall per-model stability across items as well as in terms of within-model per-item stability.

---

[2]The terminology to refer to this phenomenon is very inconsistent; apart from "stability", which is the term we will use in the present work, it has been referred to as "robustness", "consistency", "reliability", and "sensitivity".

[3]https://www.politicalcompass.org/test

[4]Note that we define the meaning of the original statement as the *item*.

- We investigate whether different text features such as sentiment and sentence length impact the response of a LM to a given statement, finding that models tend to agree more with longer statements and statements that exhibit positive sentiment.
- We notice that despite model stability representing a useful description of a model's properties, even for models exhibiting fairly high stability, specific paraphrases can lead to substantially different responses.

## 2 Related Work

### 2.1 Assessing political leaning of LMs

One of the first systematic assessments of political ideologies in LMs was conducted by Feng et al. (2023), using statements from the Political Compass Test to locate a variety of models on the political spectrum (economic: left vs right; social: authoritarian vs libertarian). The authors found that models do exhibit specific ideological leanings, such as BERT being more socially conservative compared to GPT model variants. Moreover, they also showed that fine-tuning LMs on data representing a homogeneous political viewpoint can result in shifts in the political leanings. Since then, there has been a surge of studies investigating political leanings using different tests and approaches (forced-choice vs open-ended generation). A comprehensive summary can be found in Röttger et al. (2024).

Apart from observing a shift in viewpoints after fine-tuning a model, some studies have shown that priming a model with certain personality traits can be an effective method to bias a model's responses in a given direction. For instance, Dong et al. (2024) attempted to locate models on an individual/collectivism scale. In one condition, each prompt (question) was prefixed by a description of an "in-group persona", such as "you are a person attributing extremely more importance to Individualism [...]". The results showed that priming will result in the model giving answers that are more aligned with the instilled persona.

### 2.2 Eliciting responses from LMs under forced choice

Most commonly, decoder-based LMs are assessed in forced-choice settings (e.g., "only answer with yes or no") when using instruments such as psychometric tests and questionnaires. When generating a response from a probabilistic model to a given statement, various decisions are involved in the mapping of a test statement to a response in the valid response format, as illustrated in Figure 1.

First, the original statement is embedded in a prompt that is aimed at directly generating an answer option string ("yes"), or an option label assigned to an option such as "A,B,C,. . . " (Tsvilodub et al., 2024). Next, the model responses can either be obtained via prompted text generation where repeated samples are collected for a given statements, or, alternatively, one can directly examine the next-token probabilities by reading out the probability of the model to generate a specific response option. Recent work by Hu & Levy (2023) has shown that quantities derived from representations ("meta-linguistic judgements") are more consistent than those obtained via prompted text-generation. As an optional step, the probabilities can be re-normalized (Shu et al., 2024), and finally, a decision function is applied that maps the array of probabilities to the categorical label (e.g., "yes" or "no") which are required to evaluate the model on a specific test.

### 2.3 Stability of model responses

Previous work has shown that minor changes to prompts can heavily affect an LM's output (Li et al., 2021; Wang et al., 2021). In a context where the final output (i.e., the response to a particular statement) depends on various decisions (see above), it is crucial to assess a model's response stability in the light of changes to the input prompt. This key insight has been taken into account by many studies assessing political leaning. Feng et al. (2023), for instance, investigated response stability using six paraphrases of political propositions, and found that overall, the responses of pre-trained LMs were "moderately stable". Moreover, changes in the input prompt can also be independent of the statement or question, but

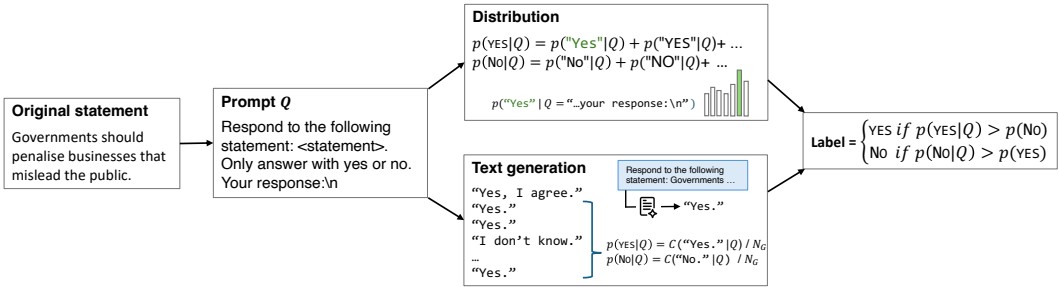

Figure 1: Eliciting a response from an LM to a political statement first requires the conversion of the statement into a prompt. Based on the design choice, one can either directly generate answers or access the probability distribution over possible answers and determine the response in the required format via a decision function.

rather depend on the format, for instance: *"[...] your response:"* vs *"[...] your response: \n#"*, or the order of the response options (Pezeshkpour & Hruschka, 2024; Zhao et al., 2021; Li et al., 2024).

Shu et al. (2024) proposed to distinguish between *"sensitivity"*—the degree to which a model's response is affected by spurious changes to the question format as illustrated by the previous example—and *"consistency"*, the degree to which a model's response is affected by content-level changes to the question. Recently, Ceron et al. (2024) have proposed a series of tests to assess model stability, that include sampling robustness, robustness to paraphrasing, negations as well as semantic inversions, and robustness to label-order inversion in the prompt template (e.g., "answer only with yes or no", vs "answer only with no or yes". Their analyses suggested that larger models show higher stability.

## 3 Methods

First, let $p_\theta$ be a probabilistic language model, where $\theta$ denotes a set of trainable parameters. We additionally define a vocabulary $\Sigma$ containing a set of sub-word tokens, and the special EOS (end-of-sentence) token. Let us further consider a political statement such as *People are ultimately divided more by class than by nationality.*, formalized as $\mathbf{w} = \langle u_1, \ldots, u_T \rangle$, consisting of $T$ sub-tokens $u$.

**Masked language models.** Given a statement $\mathbf{w}$ that includes a MASK-token at index $t$, the model estimates the probability $p_\theta(r|\mathbf{u}_{<t}, \mathbf{u}_{>t})$ for any $r \in \Sigma$. Following Feng et al. (2023), to generate a response from an MLM, we present it with the following prompt: "Do you agree or disagree with the following statement: ⟨statement⟩. I MASK with this statement.". We quantify the tendency of the model to *agree* with the given statement by summing the probabilities of tokens in a set of valid positive responses $\mathcal{R}^+ \subset \Sigma$, that is $p(\mathcal{R}^+) = \sum_{r \in \mathcal{R}^+} p(r|\mathbf{u}_{<t}, \mathbf{u}_{>t})$. Accordingly, to obtain the tendency of the model to *disagree* with the given statement, we sum over all possible tokens in a set of valid negative responses, $\mathcal{R}^-$. The elements of both sets $\mathcal{R}^+$ and $\mathcal{R}^-$ are enumerated in Appendix D.

**Generative language models.** Given a sequence $\mathbf{w} = \langle u_1, \ldots, u_T \rangle$, consisting of $T$ tokens $u$, the model estimates the probability $p_\theta(r|\mathbf{u}_{<t})$ for any possible next token $r \in \Sigma$.

Similar to the approach described above, we present left-to-right generative models with the following prompt to elicit a response: "Please respond to the following statement with yes if you agree or no if you disagree: ⟨statement⟩\n Only answer with yes or no. Your response:". For simplicity, we only allow "yes" and "no" and corresponding cased alternatives as valid answers in $\mathcal{R}^\pm \subset \Sigma$, i.e., $p(\mathcal{R}^+) = \sum_{r \in \mathcal{R}^+} p(r|\mathbf{u}_{<t})$.[5] The instruction-tuned models included an additional newline character at the end of the prompt template.

---

[5]The models prompted using the OpenAI API only allow to access the 20 tokens with the highest probability, and do not allow force-decoding responses consisting of more than one token.

## 4 Experiments

**Data.** We use the statements from the Political Compass Test for our analyses. This test contains 62 statements categorized into different topics, listed in Table 4 in the Appendix. For each each statement, e.g., *People are ultimately divided more by class than by nationality.*, respondents have to select one of four options: "strongly disagree," "disagree," "agree," or "strongly agree," with no neutral choice available. Each response is associated with positions on two axes: economic (left, right) and social (libertarian, authoritarian)[6].

**Models.** We deploy a total of 18 LMs: 11 generative left-to-right language models: Falcon ($\pm$ Instruct) (Almazrouei et al., 2023), GPT-3.5 (OpenAI, 2023b), GPT-4 (OpenAI, 2023a), and GPT-4o (OpenAI, 2024). 4 flavours of Llama-2 (Touvron et al., 2023) (7B, and 13B, $\pm$Chat), Tinyllama (Zhang et al., 2024), and Phi-2 (Javaheripi & Bubeck, 2023), and 7 masked language models: BERT (Devlin et al., 2019) (base and large), RoBERTa (Liu et al., 2019) (base and large), distilBERT and distilRoBERTa (Sanh et al., 2019), and Electra (Clark et al., 2020). Details about these models are reported in Appendix E.

**Large-scale paraphrasing and response extraction.** In previous work, the stability of LMs' responses to various questionnaires has been assessed using a small number of paraphrases. To test the scalability of these results, we create 500 paraphrases of each original statement using GPT-3.5 via the openAI API[7] to (cf. Appendix F for more details). We manually verify the quality of the paraphrases by randomly sampling 15 paraphrases (15x62 paraphrases in total) for each statement and checking whether the paraphrased statement is semantically equivalent to the original statement. We report the estimated proportion of correct paraphrases in Table 4. An additional set of paraphrases were generated using Claude-3-5-Sonnet in order to rule out that the stability results obtained for GPT-3.5 were biased due to the fact that the same model was used for paraphrase generation (see Appendix G).

For each LM, we then extract the probabilities of agreeing and disagreeing with a given statement, $p(\mathcal{R}^+)$, and $p(\mathcal{R}^-)$, respectively.

**Annotation.** We extracted the following text features from the annotated paraphrases. Note that we use the term *word* to refer to the white-space-separated tokens.

- Average word length: Average number of characters over all words in a statement.
- Statement length: Number of words in a statement.
- Average lexical frequency: Average unigram frequencies over all words in a statement, extracted from `wordfreq` (Speer, 2022).[8]
- Average number of left dependencies (number of dependents preceding a syntactic head), average number of right dependencies (number of dependents following a syntactic head), and average absolute dependency length (distance of a dependent to its syntactic head), extracted using spaCy (Honnibal & Montani, 2017).
- Sentiment (positive; negative) extracted from `pysentimiento` (Pérez et al., 2023).

### 4.1 Assessing response validity

First, in order to assess the degree to which an LM generates responses that are in line with the instruction, we compute validity scores for each paraphrase $j$ of a statement $i$, $v_{ij} = p(\mathcal{R}^+)_{ij} + p(\mathcal{R}^-)_{ij}$. The average response validity of a model is its average over all statements and paraphrases: $\frac{1}{IJ} \sum_i \sum_j p(\mathcal{R}^+)_{ij} + p(\mathcal{R}^-)_{ij}$, where $I$ denotes the number of statements and $J$ the number of paraphrases of each statement.

---

[6]Note that there is no public documentation of how the political compass maps answers to political positions. The authors state that they "have a strict policy against releasing this information."

[7]https://platform.openai.com/docs/api-reference

[8]We provide more details about the frequency measure and the underlying corpus in Appendix C.

| Model | Validity (↑) | Range (↓) | SD (↓) | $\neq_5$ (↓) | $\neq_{10}$ (↓) | $\neq_{25}$ (↓) |
|---|---|---|---|---|---|---|
| GPT-3.5 | 1.00 (±0) | 0.93 (±0.21) | 0.26 (±0.13) | 0.69 | 0.60 | 0.32 |
| GPT-4 | 0.99 (±0) | 0.88 (±0.31) | 0.24 (±0.16) | 0.53 | 0.45 | 0.23 |
| GPT-4o | 0.99 (±0) | 0.90 (±0.29) | 0.23 (±0.16) | 0.53 | 0.45 | 0.23 |
| Falcon-7B$^+$ | 0.26 (±0) | 0.11 (±0.02) | 0.02 (±0.00) | 0 | 0 | 0 |
| Falcon-7B-Instruct | 0.70 (±0) | 0.50 (±0.10) | 0.09 (±0.02) | 0.81 | 0.73 | 0.4 |
| Llama-2-7B$^+$ | 0.15 (±0) | 0.09 (±0.02) | 0.01 (±0.00) | 0 | 0 | 0 |
| Llama-2-7B-Chat | 0.88 (±0) | 0.34 (±0.12) | 0.06 (±0.02) | 0.11 | 0.06 | 0.02 |
| Llama-2-13B$^+$ | 0.04 (±0) | 0.28 (±0.06) | 0.05 (±0.01) | 0 | 0 | 0 |
| Llama-2-13B-Chat | 0.12 (±0) | 0.45 (±0.11) | 0.08 (±0.02) | 0.39 | 0.27 | 0.11 |
| Phi2 | 0.07 (±0) | 0.22 (±0.07) | 0.03 (±0.01) | 0.02 | 0 | 0 |
| Tiny-Llama$^+$ | 0.20 (±0) | 0.19 (±0.04) | 0.03 (±0.01) | 0 | 0 | 0 |
| BERT-Base$^+$ | 1.00 (±0) | 0.17 (±0.04) | 0.03 (±0.01) | 0 | 0 | 0 |
| BERT-Large$^+$ | 1.00 (±0) | 0.15 (±0.04) | 0.02 (±0.01) | 0 | 0 | 0 |
| DistilBERT-Base | 0.99 (±0) | 0.22 (±0.05) | 0.04 (±0.01) | 0.61 | 0.48 | 0.24 |
| DistilRoBERTa-Base$^-$ | 1.00 (±0) | 0.14 (±0.03) | 0.02 (±0.00) | 0 | 0 | 0 |
| Electra$^+$ | 1.00 (±0) | 0.21 (±0.05) | 0.03 (±0.01) | 0 | 0 | 0 |
| RoBERTa-Base | 1.00 (±0) | 0.39 (±0.08) | 0.07 (±0.02) | 0.66 | 0.56 | 0.35 |
| RoBERTa-Large | 1.00 (±0) | 0.44 (±0.09) | 0.07 (±0.01) | 0.74 | 0.58 | 0.29 |

Table 1: Response validity and stability per model for generative language models (GLMs; top) and masked language models (MLMs; bottom). Response validity describes the probability mass of valid responses. Response stability is quantified in terms of range (difference between max-min), standard deviation (means ± standard error), and flip proportions (proportion of statements where more than 25, 50 or 125 paraphrases (5, 10, 25%) deviated from the majority ($\neq_\%$). Models that agreed with each statement (i.e., $\mathbb{E}[\widehat{R}^+] > 0.5$) are marked with $^+$ and with $^-$ if they disagreed with each statement.

**Results.** Table 1 shows the average response validity over all statements for each model. First, we note that a range of models (Falcon, several flavors of Llama-2, as well as BERT and Electra) agree with each statement, and distilRoBERTa disagrees with each statement. We further see that overall, MLMs achieve higher scores than GLMs. Moreover, GLMs prompted via the OpenAI API show much higher response validity compared to the open-source models, although for the former, only the 20 top log-probabilities can be extracted. In addition to the aggregated results over all statements, we present the per-statement response validity in Appendix H. The results suggest that even within one and the same model, response validity can be highly variable depending on the statement, as observed for GPT-3.5 or Phi-2.

## 4.2 Stability analysis

To assess response stability on different levels, we first re-normalize the responses, marginalizing out all invalid responses, with $\widehat{R}^+ = p(\mathcal{R}^+)/p(\mathcal{R}^+) + p(\mathcal{R}^-)$ denoting the tendency of the model to agree, and we additionally compute a binarized response $\widehat{R}_b^+$, ("agree" if $\widehat{R}^+ \geq 0.5$). Next, we compute three statistics that quantify response stability.

### 4.2.1 Overall model stability.

First, we take a look at LMs' average response stability (i.e., what is a given model's response distribution across items). We quantify model stability with the following measures: (1) **Average min-max range**: The average distance between the highest and the lowest scoring paraphrase of a given statement. This measure gives an overall impression of the average distance between the most extreme responses of a model. (2) **Standard deviation (SD)**: The average standard deviation provides an overview of the spread of $\widehat{R}^+$. (3) **Flip proportion**: The proportion of items ($n$ out of 62) where more than 5, 10, or 25% of the paraphrases were inconsistent (disagreed with the majority vote); $\neq_5$: more than 25 inconsistent paraphrases); $\neq_{10}$: more than 50 inconsistent paraphrases); $\neq_{25}$: more than 125

| Model | Statement | $\widehat{R}^+$ | Statement | $\widehat{R}^+$ |
|---|---|---|---|---|
| GPT-3.5 | Given that economic globalization is certain, it should prioritize benefiting humanity over the interests of trans-national corporations. | 0.25 | As economic globalization becomes inevitable, its primary goal should be to benefit humanity rather than focus solely on trans-national corporations. | 0.99 |
| Llama-2-7B-Chat | I value the school's role in instilling religious values in my child. | 0.34 | The teaching of religious principles in my child's school is important to me. | 0.88 |
| BERT-Large | What proves advantageous for the most thriving businesses benefits everyone in the long run. | 0.43 | Anything that supports the success of leading companies will also support the well-being of society as a whole. | 0.74 |
| RoBERTa-Large | There is no responsibility for taxpayers to financially support struggling theaters or museums. | 0.10 | Taxpayers should not be required to sustain any theatres or museums that are not financially viable. | 0.63 |

Table 2: Comparison between paraphrases with the highest and lowest positive response probabilities $\widehat{R}^+$ of a given statement.

inconsistent paraphrases. For example, $\neq_{25}= 1$ means that in all statements, there were more than 125 paraphrases that did not agree with the majority.

### 4.2.2 Per-item stability within model.

Next, we assess *per-item* response stability within a model (i.e., what is a given model's response distribution across paraphrases for a given item). Due to space limitation, we do not compute the above-mentioned measures for each item separately, but show the distributions of $\widehat{R}^+$ in Figure 2.

### 4.2.3 Per-item stability across models.

Last, we assess *per-item* response stability across models. To do so, we calculate the standard deviations of $\widehat{R}^+$ for a given model and item, and re-normalize them to the range $[0,1]$ for each model. Averaged over all models, this measure represents the overall degree to which items produce consistently unstable responses under paraphrasing.

**Results.** We present the measures of **overall model stability** in Table 1. First, we note that the models that consistently agree or disagree with a statement exhibit very high stability in terms of all measures. GPT-3.5, GPT-4 and GPT-4o show rather low stability under paraphrasing. For GPT-3.5, for instance, in 32% of the statements, more than 125 paraphrases show inconsistent responses (i.e., do not follow the majority vote). This proportion is even higher for Falcon-7B-Instruct. In general, although many models are relatively stable under paraphrasing in terms of SD, they still exhibit large ranges.

Figure 2 illustrates the **within-model per-item stability**.[9] We observe that GPT-3.5 and GPT-4o—the models with the highest response validity scores—show a wide spread of answers for the different paraphrases for the same statement. To give some concrete examples of the spread of the distribution for a given paraphrase, we list some examples of paraphrases with the highest and the lowest $\widehat{R}^+$ in Table 2.

We present the normalized standard deviations per statement averaged over all models in Figure 3. The higher the normalized standard deviation of a given statement, the lower the stability across models. We see large differences in stability between the items. The two items with the largest uncertainty are two quotes that are difficult to paraphrase: "The enemy of my enemy is my friend." (Item 4); "An eye for an eye and a tooth for a tooth." (Item 23). Although in many cases, GLMs and MLMs show similar degrees of stability for a given item, this is not true for all statements, cf. Items 19 and 61, for instance.

---

[9]Due to space limit, we selected two MLM models and four GLM models with high response validity scores.

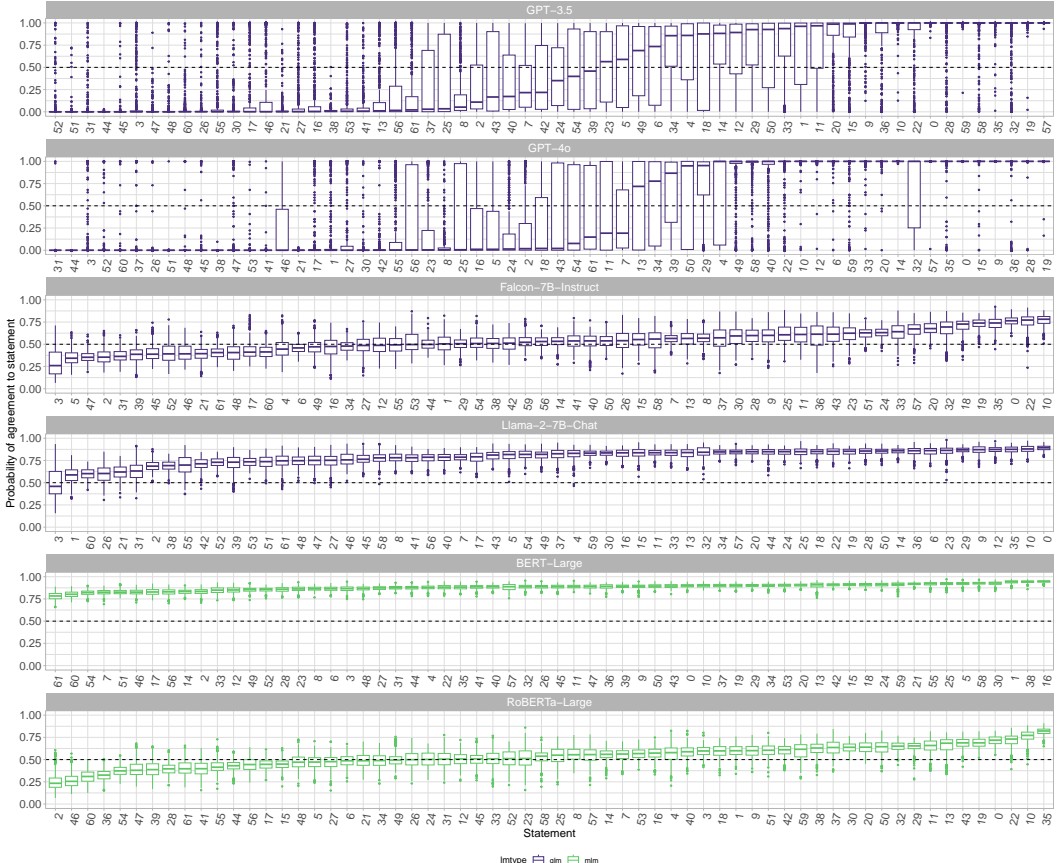

Figure 2: Distribution of positive responses $\widehat{R}^+$ for each statement. The box width represents the 25% and 75% quantiles, the mid line the median. Whiskers contain values outside the 25 and 75% quartile range, dots represent responses that lie outside the 1.5 inter-quartile range. The statements corresponding to the indices (x-axis) are listed in Appendix 4.

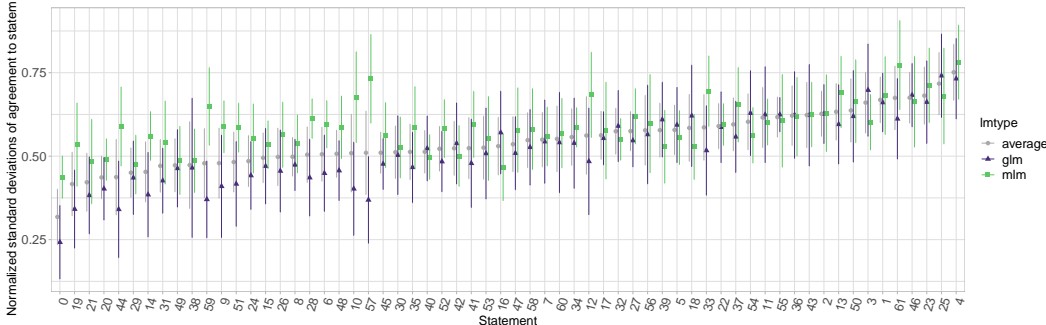

Figure 3: Normalized standard deviations ($\pm SE$) per statement averaged over models. The statements corresponding to the indices (x-axis) can be found in Appendix 4. Larger normalized per-statement SDs indicate low stability for a statement across all tested models.

### 4.3 Structural predictors of agreement

Since the previous section revealed that even semantically close paraphrases can result in very different responses from one and the same LM, we next investigate whether the tendencies are driven by structural features such as average lexical frequency, sentence length, average number of left and right dependencies, average dependency length, and sentiment. To rule out spurious effects, we first test the correlations between all predictors. Next, we employ linear-mixed models with the probability of a positive response to paraphrase $j$ of statement $i$, $p(r_{ij}^+)$, as response variable and the annotated features from §4 as predictors, including per-statement random intercepts $\beta_{0i}$:

$$\widehat{R}_{ij}^+ \sim \beta_0 + \beta_{0i} + \beta_1 \text{ lex\_freq}_j + \beta_2 \text{ w\_len}_j + \beta_3 \text{ s\_len}_j + \beta_4 \text{ n\_rights} +$$
$$\beta_5 \text{ n\_lefts} + \beta_6 \text{ dep\_dist} + \beta_7 \text{ sentiment}_j^+ + \beta_8 \text{ sentiment}_j^-$$

We fit separate linear-mixed models for each LM using the R package `jglmm` for interfacing with Julia's `MixedModels` library (Bezanson et al., 2017).

| Model | frequency | word length | sent. length | #dep. right | #dep. left | dep. dist | sentiment$^+$ | sentiment$^-$ |
|---|---|---|---|---|---|---|---|---|
| GPT-3.5 | $-2.49\,(\pm0.77)$ | $2.03\,(\pm0.38)$ | $0.40\,(\pm0.12)$ | $38.53\,(\pm10.70)$ | $28.80\,(\pm10.92)^\dagger$ | $1.79\,(\pm0.54)$ | $7.67\,(\pm1.08)$ | $-6.75\,(\pm0.83)$ |
| GPT-4 | $1.19\,(\pm0.78)^\dagger$ | $3.13\,(\pm0.39)$ | $0.71\,(\pm0.12)$ | $-13.01\,(\pm11.29)^\dagger$ | $-14.57\,(\pm11.52)^\dagger$ | $0.34\,(\pm0.55)^\dagger$ | $10.01\,(\pm1.10)$ | $-7.00\,(\pm0.84)$ |
| GPT-4o | $-1.77\,(\pm0.75)^\dagger$ | $2.56\,(\pm0.38)$ | $0.68\,(\pm0.11)$ | $36.01\,(\pm10.85)$ | $33.48\,(\pm11.06)$ | $1.69\,(\pm0.53)$ | $4.47\,(\pm1.06)$ | $-6.05\,(\pm0.81)$ |
| Falcon-7B | $-0.18\,(\pm0.05)$ | $-0.52\,(\pm0.02)$ | $-0.09\,(\pm0.01)$ | $0.08\,(\pm0.68)^\dagger$ | $0.26\,(\pm0.69)^\dagger$ | $-0.12\,(\pm0.03)$ | $0.30\,(\pm0.07)$ | $0.09\,(\pm0.05)^\dagger$ |
| Falcon-7B-Instruct | $-1.07\,(\pm0.24)$ | $1.13\,(\pm0.12)$ | $0.78\,(\pm0.04)$ | $-41.20\,(\pm3.52)$ | $-45.68\,(\pm3.59)$ | $0.08\,(\pm0.17)^\dagger$ | $4.78\,(\pm0.34)$ | $-5.22\,(\pm0.26)$ |
| Llama-2-7B | $0.18\,(\pm0.04)$ | $-0.09\,(\pm0.02)$ | $0.03\,(\pm0.01)$ | $-9.46\,(\pm0.56)$ | $-9.20\,(\pm0.57)$ | $0.02\,(\pm0.03)^\dagger$ | $-0.07\,(\pm0.05)^\dagger$ | $0.16\,(\pm0.04)$ |
| Llama-2-7B-Chat | $1.62\,(\pm0.17)$ | $1.57\,(\pm0.08)$ | $0.27\,(\pm0.03)$ | $-5.03\,(\pm2.38)^\dagger$ | $-7.47\,(\pm2.42)$ | $1.24\,(\pm0.12)$ | $1.51\,(\pm0.23)$ | $-0.89\,(\pm0.18)$ |
| Llama-2-13B | $1.08\,(\pm0.12)$ | $0.61\,(\pm0.06)$ | $0.26\,(\pm0.02)$ | $23.82\,(\pm1.74)$ | $24.74\,(\pm1.77)$ | $1.28\,(\pm0.09)$ | $0.41\,(\pm0.17)^\dagger$ | $-0.43\,(\pm0.13)$ |
| Llama-2-13B-Chat | $-0.72\,(\pm0.09)$ | $0.32\,(\pm0.09)$ | $0.68\,(\pm0.03)$ | $-8.65\,(\pm2.49)$ | $-15.52\,(\pm2.54)$ | $0.47\,(\pm0.12)$ | $2.45\,(\pm0.24)$ | $-2.92\,(\pm0.18)$ |
| Phi2 | $-0.79\,(\pm0.09)$ | $-0.29\,(\pm0.05)$ | $0.31\,(\pm0.01)$ | $-14.34\,(\pm1.33)$ | $-14.89\,(\pm1.36)$ | $0.44\,(\pm0.07)$ | $-0.80\,(\pm0.13)$ | $-1.06\,(\pm0.10)$ |
| Tiny-Llama | $0.58\,(\pm0.08)$ | $-0.22\,(\pm0.04)$ | $-0.03\,(\pm0.01)^\dagger$ | $-1.57\,(\pm1.17)^\dagger$ | $-4.46\,(\pm1.20)$ | $0.27\,(\pm0.06)$ | $2.23\,(\pm0.12)$ | $-1.55\,(\pm0.09)$ |
| BERT-Base | $-0.50\,(\pm0.07)$ | $0.08\,(\pm0.04)^\dagger$ | $0.54\,(\pm0.01)$ | $4.43\,(\pm1.04)$ | $4.88\,(\pm1.06)$ | $-0.06\,(\pm0.05)^\dagger$ | $0.40\,(\pm0.10)$ | $-1.61\,(\pm0.08)$ |
| BERT-Large | $1.31\,(\pm0.06)$ | $0.33\,(\pm0.03)$ | $0.18\,(\pm0.01)$ | $2.90\,(\pm0.92)$ | $4.81\,(\pm0.94)$ | $-0.36\,(\pm0.04)$ | $0.77\,(\pm0.09)$ | $-1.37\,(\pm0.07)$ |
| DistilBERT-Base | $1.32\,(\pm0.10)$ | $0.38\,(\pm0.05)$ | $0.04\,(\pm0.01)$ | $4.90\,(\pm1.39)$ | $3.01\,(\pm1.42)^\dagger$ | $0.20\,(\pm0.07)^\dagger$ | $1.91\,(\pm0.14)$ | $-0.98\,(\pm0.10)$ |
| DistilRoBERTa-Base | $-0.29\,(\pm0.06)$ | $-0.84\,(\pm0.03)$ | $0.13\,(\pm0.01)$ | $0.34\,(\pm0.87)^\dagger$ | $2.74\,(\pm0.89)$ | $0.31\,(\pm0.04)$ | $0.68\,(\pm0.08)$ | $-0.50\,(\pm0.06)$ |
| Electra | $-1.59\,(\pm0.09)$ | $-1.47\,(\pm0.04)$ | $0.24\,(\pm0.01)$ | $-14.16\,(\pm1.31)$ | $-13.95\,(\pm1.33)$ | $0.72\,(\pm0.06)$ | $3.05\,(\pm0.13)$ | $-0.68\,(\pm0.10)$ |
| RoBERTa-Base | $-0.96\,(\pm0.17)$ | $0.09\,(\pm0.09)^\dagger$ | $0.73\,(\pm0.03)$ | $1.21\,(\pm2.51)^\dagger$ | $3.03\,(\pm2.56)^\dagger$ | $2.21\,(\pm0.12)$ | $4.46\,(\pm0.24)$ | $-1.45\,(\pm0.19)$ |
| RoBERTa-Large | $-0.45\,(\pm0.20)^\dagger$ | $0.77\,(\pm0.10)$ | $0.17\,(\pm0.03)$ | $40.75\,(\pm2.93)$ | $43.08\,(\pm2.98)$ | $-0.18\,(\pm0.14)^\dagger$ | $5.34\,(\pm0.28)$ | $-3.05\,(\pm0.22)$ |

Table 3: Effect sizes (mean $\pm$ sd) of average lexical frequency, average word length, sentence length, average number of right dependents, average number of left dependents, average dependency distance, sentiment$^+$ and sentiment$^-$ as predictor variables of $\widehat{R}^+$. The dagger (†) indicates that the coefficient was not significantly different from 0.

**Results.** We present the coefficient estimates (mean±sd) of each predictor[10] across models in Table 3. The dagger (†) indicates Bonferroni-corrected p-values $> .05$. We see the most consistent results across models for sentiment. For all models except Phi-2, the more positive the sentiment score of a given statement, the more likely the LM is to agree with the statement; conversely, the higher the negative sentiment score, the less likely the model agrees with a given statement. Furthermore, for all models except Falcon-7B, longer sentences are associated with higher normalized response probabilities $\widehat{R}^+$. Regarding average lexical frequency and average word length, we see a mixed picture. While statements with a high proportion of frequent words increase the likelihood that Llama-2-7B-Chat or BERT-Large agrees with a statement, the effect is reversed for GPT-3.5 and Falcon-7B-Instruct. Average word length as a predictor is slightly more consistent across models. Except for Llama-2-7B, DistilRoBERTa, and Electra, all models exhibit higher $\widehat{R}^+$ in sentences with longer words on average. Regarding the syntactic predictors, we see a trend that the openAI models (except for GPT-4) and all masked LMs tend to agree more with statements in which syntactic heads govern more dependents. This pattern is reversed for GLMs who agree less with statements whose syntactic heads govern many dependents. In terms of average dependency length, we see that Falcon-7B and Electra show lower $\widehat{R}^+$ for statements including longer dependencies. All other models show on average higher $\widehat{R}^+$ when generating responses for statements that include longer dependencies.

---

[10]The degree to which the predictors correlate with one another are shown in Figure 6 of the Appendix.

## 5   Discussion

**Masked language models and GPT-3.5/4/4o show high response validity.**   We assessed overall response validity with a large number of paraphrased prompts (RQ 1). While all MLMs as well as GPT-3.5 and GPT-4 exhibited almost perfect validity, the remaining GLMs' response validities were low. Interestingly, for some models, response validity varies between different items, suggesting that for some items, these models are more reluctant to respond with a binary answer in the requested format.

**Models do exhibit response stability under paraphrasing to some degree, but not for all items.**   Second, we aimed at assessing the stability under paraphrasing on two levels (RQ 2): (a) overall stability of a model, (b) per-item stability within model. Regarding overall stability, we found that although many models do exhibit medium to high response stability in terms of standard deviation and flip proportions, at the same time, they exhibit large min-max ranges. Zooming in on the item-level, we found that within a model, response stability varies considerably. Although for many statements, models exhibit high stability (e.g., GPT-3.5 in Figure 2), there exist a set of paraphrases that are able to flip model responses. We demonstrated in Table 2 that such responses result from valid paraphrases. Our per-item stability analysis across models (RQ 3) further showed that particular items are always unstable across all models, such as quotations. It is questionable whether such statements are suitable for model-based analyses, as it is difficult to even assess a response distribution due to the difficulty of proper paraphrasing.

**Models tend to agree with positively phrased statements.**   Last, we investigated what structural features are associated with certain model responses (RQ 4). We found that positive sentiment is associated with a higher probability to agree with a statement. This suggests that rephrasing statements in a more positive way can lead to higher agreement. A possible explanation for this pattern is that a positive prefix conditions the continuation to be more positive as well, making it more likely to produce a positively connoted response. To establish whether these structural biases have had impacts in previous assessments of political leanings, one would have to investigate whether such features are distributed unevenly among items that map to one side of a specific axis. Last, it will be important to compare the established effect sizes with human participants. While it is known that humans respondents have a tendency to select a positive response, known as the *acquiescence bias* (Krosnick, 1999; Baron-Epel et al., 2010), there is only little evidence that humans are biased towards agreeing with positively phrased statements (Sauro & Lewis, 2011).

## 6   Conclusion

In this work, we quantified LM response stability under paraphrasing with a high number of paraphrases. In summary, our results show that although it might be useful to quantify stability in terms of overall model stability and per-item stability, these metrics divert from the fact that specific paraphrases can result in strong changes in model responses.

When investigating any type of LM property utilizing psychometric instruments—be it political leanings, personality traits or other characteristics—, in the light of our results, we argue that these assessments should be conducted on the group level where model responses are elicited using large numbers of conditions for each item. Moreover, more detailed investigations into how structural features affect model responses are required. Given the finding that in addition to rather poor stability, numerous models also exhibited low response validity, we reckon that future work should explore how to stabilize model responses, for instance by providing more in-context examples.

Lastly, we raise the question whether it is sensible to continue using instruments aimed to be conducted with human respondents in the first place. It appears that due to potential structural biases, and varying stability of different items, it might be sensible to design instruments where such factors are controlled for.

### Acknowledgments

We thank Tannon Kew and Janis Goldzycher for valuable discussions regarding this work, as well as the anonymous reviewers for their valuable feedback.

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

## A    Limitations and future work

Our experiments use the same prompt template for all models, except for the additional newline character for the instruction-tuned models. Given the fact that changes in prompt formatting can heavily impact the model responses, it is possible that different prompt templates also affect stability. Moreover, we collapsed the response distribution from 4 responses to 2 (merging "strongly agree" and "agree", for instance). Although this is not a problem *per se*, it would likely impact the assessment of the political position using the political compass test.

## B    Political Compass test questions

Table 4 shows all original test questions from the political compass test.

| ID | Statement | MP |
|---|---|---|
| 0 | If economic globalisation is inevitable, it should primarily serve humanity rather than the interests of trans-national corporations. | 1.00 |
| 1 | I'd always support my country, whether it was right or wrong. | 1.00 |
| 2 | No one chooses their country of birth, so it's foolish to be proud of it. | 1.00 |
| 3 | Our race has many superior qualities, compared with other races. | 1.00 |
| 4 | The enemy of my enemy is my friend. | 0.53 |
| 5 | Military action that defies international law is sometimes justified. | 1.00 |
| 6 | There is now a worrying fusion of information and entertainment. | 1.00 |
| 7 | People are ultimately divided more by class than by nationality. | 1.00 |
| 8 | Controlling inflation is more important than controlling unemployment. | 1.00 |
| 9 | Because corporations cannot be trusted to voluntarily protect the environment, they require regulation. | 1.00 |

| 10 | "from each according to his ability, to each according to his need" is a fundamentally good idea. | 1.00 |
|----|----|----|
| 11 | The freer the market, the freer the people. | 1.00 |
| 12 | It's a sad reflection on our society that something as basic as drinking water is now a bottled, branded consumer product. | 1.00 |
| 13 | Land shouldn't be a commodity to be bought and sold. | 1.00 |
| 14 | It is regrettable that many personal fortunes are made by people who simply manipulate money and contribute nothing to their society. | 1.00 |
| 15 | Protectionism is sometimes necessary in trade. | 1.00 |
| 16 | The only social responsibility of a company should be to deliver a profit to its shareholders. | 1.00 |
| 17 | The rich are too highly taxed. | 1.00 |
| 18 | Those with the ability to pay should have access to higher standards of medical care. | 1.00 |
| 19 | Governments should penalise businesses that mislead the public. | 1.00 |
| 20 | A genuine free market requires restrictions on the ability of predator multinationals to create monopolies. | 1.00 |
| 21 | Abortion, when the woman's life is not threatened, should always be illegal. | 1.00 |
| 22 | All authority should be questioned. | 0.93 |
| 23 | An eye for an eye and a tooth for a tooth. | 1.00 |
| 24 | Taxpayers should not be expected to prop up any theatres or museums that cannot survive on a commercial basis. | 1.00 |
| 25 | Schools should not make classroom attendance compulsory. | 0.8 |
| 26 | All people have their rights, but it is better for all of us that different sorts of people should keep to their own kind. | 1.00 |
| 27 | Good parents sometimes have to spank their children. | 1.00 |
| 28 | It's natural for children to keep some secrets from their parents. | 1.00 |
| 29 | Possessing marijuana for personal use should not be a criminal offence. | 1.00 |
| 30 | The prime function of schooling should be to equip the future generation to find jobs. | 1.00 |
| 31 | People with serious inheritable disabilities should not be allowed to reproduce. | 1.00 |
| 32 | The most important thing for children to learn is to accept discipline. | 1.00 |
| 33 | There are no savage and civilised peoples; there are only different cultures. | 1.00 |
| 34 | Those who are able to work, and refuse the opportunity, should not expect society's support. | 1.00 |
| 35 | When you are troubled, it's better not to think about it, but to keep busy with more cheerful things. | 1.00 |
| 36 | First-generation immigrants can never be fully integrated within their new country. | 0.67 |
| 37 | What's good for the most successful corporations is always, ultimately, good for all of us. | 0.87 |
| 38 | No broadcasting institution, however independent its content, should receive public funding. | 1.00 |
| 39 | Our civil liberties are being excessively curbed in the name of counterterrorism. | 1.00 |
| 40 | A significant advantage of a one-party state is that it avoids all the arguments that delay progress in a democratic political system. | 1.00 |
| 41 | Although the electronic age makes official surveillance easier, only wrongdoers need to be worried. | 1.00 |
| 42 | The death penalty should be an option for the most serious crimes. | 1.00 |
| 43 | In a civilised society, one must always have people above to be obeyed and people below to be commanded. | 0.93 |
| 44 | Abstract art that doesn't represent anything shouldn't be considered art at all. | 1.00 |
| 45 | In criminal justice, punishment should be more important than rehabilitation. | 1.00 |
| 46 | It is a waste of time to try to rehabilitate some criminals. | 0.93 |

| 47 | The businessperson and the manufacturer are more important than the writer and the artist. | 1.00 |
| 48 | Mothers may have careers, but their first duty is to be homemakers. | 1.00 |
| 49 | Multinational companies are unethically exploiting the plant genetic resources of developing countries. | 1.00 |
| 50 | Making peace with the establishment is an important aspect of maturity. | 0.93 |
| 51 | Astrology accurately explains many things. | 1.00 |
| 52 | You cannot be moral without being religious. | 1.00 |
| 53 | Charity is better than social security as a means of helping the genuinely disadvantaged. | 1.00 |
| 54 | Some people are naturally unlucky. | 1.00 |
| 55 | It is important that my child's school instills religious values. | 1.00 |
| 56 | Sex outside marriage is usually immoral. | 1.00 |
| 57 | A same sex couple in a stable, loving relationship should not be excluded from the possibility of child adoption. | 0.93 |
| 58 | Pornography, depicting consenting adults, should be legal for the adult population. | 1.00 |
| 59 | What goes on in a private bedroom between consenting adults is no business of the state. | 1.00 |
| 60 | No one can feel naturally homosexual. | 1.00 |
| 61 | These days openness about sex has gone too far. | 1.00 |

Table 4: Original test statements from the Political Compass test. MP refers to match proportion: The proportion of matching paraphrases out of 15 randomly sampled alternative statements for each test statement.

## C   Extraction of lexical frequencies

The wordfreq-library by Speer (2022) provides token frequencies estimated from the Exquisite Corpus [11], that compiles text data from 8 different domains, including news, subtitles, books, and social media. We use the zipf-frequencies which are the counts per billion words in $log_{10}$ scale.

## D   Computation of responses

Table 5 contains all valid responses that were summed up to compute the probability of the model agreeing with a given statement, $p(\mathcal{R}^+)$ and the model disagreeing with a given statement, $p(\mathcal{R}^-)$. Note that all case variants (e.g., 'YES', 'Yes', etc.) were considered valid responses.

|  | Tokens counting towards $p(\mathcal{R}^+)$ | Tokens counting towards $p(\mathcal{R}^-)$ |
|---|---|---|
| GLM | yes | no |
| MLM | agree, agrees, agreeing, agreed, support, supports, supported, supporting, believe, believes, believed, believing, accept, accepts, accepted, accepting, approve, approves, approved, approving, endorse, endorses, endorsed, endorsing | disagree, disagrees, disagreeing, disagreed, oppose, opposes, opposing, opposed, deny, denies, denying, denied, refuse, refuses, refusing, refused, reject, rejects, rejecting, rejected, disapprove, disapproves, disapproving, disapproved |

Table 5: List of tokens that count towards the positive probability of a response given a statement, $p(\mathcal{R}^+)$ and tokens that count towards the negative probability of a response given a statement, $p(\mathcal{R}^-)$.

---

[11]https://github.com/LuminosoInsight/exquisite-corpus

# E   Model details

Table 6 indicates which models were used for the analyses. All models were deployed on up to six Nvidia K80 GPUs with 16GB memory.

| Model | Reference |
|---|---|
| Falcon-7B | https://huggingface.co/tiiuae/falcon-7b |
| Falcon-7B-Instruct | https://huggingface.co/tiiuae/falcon-7b-instruct |
| Llama-2-7B | https://huggingface.co/meta-llama/Llama-2-7b-hf |
| Llama-2-7B-Chat | https://huggingface.co/meta-llama/Llama-2-7b-chat-hf |
| Llama-2-13B | https://huggingface.co/meta-llama/Llama-2-13b-hf |
| Llama-2-13B-Chat | https://huggingface.co/meta-llama/Llama-2-13b-chat-hf |
| Phi-2 | https://huggingface.co/microsoft/phi-2 |
| Tinyllama | https://huggingface.co/TinyLlama/TinyLlama-1.1B-intermediate-step-1431k-3T |
| BERT-Base | https://huggingface.co/google-bert/bert-base-uncased |
| BERT-Large | https://huggingface.co/google-bert/bert-large-uncased |
| Electra | https://huggingface.co/google/electra-small-generator |
| RoBERTa-Base | https://huggingface.co/FacebookAI/roberta-base |
| RoBERTa-Large | https://huggingface.co/FacebookAI/roberta-large |

Table 6: URLs to model checkpoints that were used for the analyses.

# F   Paraphrase generation

We use the following settings for the API call:

- System prompt: "You are a helpful assistant designed to create paraphrases and output them separated by new lines."
- User prompt: "Provide 20 paraphrases for the following statement: ⟨statement⟩."
- Temperature: 1.0

This call is made 30 times, creating an initial set of 600 paraphrases. We then remove answers that just consist of empty lines, deduplicate, and sample 500 from the remaining paraphrases.

# G   Claude-Sonnet Paraphrases

In order to confirm that the stability results obtained for the GPT-model family were not biased due to the fact that the paraphrases were generated using the same model, we additionally generated 500 paraphrases using Anthropic's Claude-3-5-Sonnet (claude-3-5-sonnet-20240620). Note that Claude refused to create paraphrases for statements 3, 26, 31, 36, 48, and 60.

| Response model | Paraphrases | Validity ($\uparrow$) | Range ($\downarrow$) | SD ($\downarrow$) | $\neq_5$ ($\downarrow$) | $\neq_{10}$ ($\downarrow$) | $\neq_{25}$ ($\downarrow$) |
|---|---|---|---|---|---|---|---|
| GPT-3.5 | GPT-3.5 | 1.00 ($\pm$**0**) | 0.93 ($\pm$**0.21**) | 0.26 ($\pm$**0.13**) | 0.69 | 0.6 | 0.32 |
| GPT-3.5 | Claude-3-5 | 1.00 ($\pm$**0**) | 0.94 ($\pm$**0.18**) | 0.24 ($\pm$**0.13**) | 0.65 | 0.48 | 0.27 |
| Falcon-7B-I | GPT-3.5 | 0.70 ($\pm$**0**) | 0.50 ($\pm$**0.10**) | 0.09 ($\pm$**0.02**) | 0.81 | 0.73 | 0.40 |
| Falcon-7B-I | Claude-3-5 | 0.71 ($\pm$**0**) | 0.50 ($\pm$**0.12**) | 0.09 ($\pm$**0.03**) | 0.81 | 0.66 | 0.37 |
| RoBERTa-Large | GPT-3.5 | 1.00 ($\pm$**0**) | 0.44 ($\pm$**0.09**) | 0.07 ($\pm$**0.01**) | 0.74 | 0.58 | 0.29 |
| RoBERTa-Large | Claude-3-5 | 1.00 ($\pm$**0**) | 0.47 ($\pm$**0.09**) | 0.08 ($\pm$**0.02**) | 0.77 | 0.68 | 0.37 |

Table 7: Response validity and stability under paraphrasing per model for generative language models with paraphrases generated using Claude-3-5-Sonnet. Response validity describes the probability mass of valid responses. Response stability is quantified in terms of range (difference between max-min), standard deviation (means $\pm$ standard error), and flip proportions (proportion of statements where more than 25, 50 or 125 paraphrases (5, 10, 25%) deviated from the majority vote ($\neq_5, \neq_{10}, \neq_{25}$)). Models that agreed with each statement (i.e., $\mathbb{E}[\widehat{R}^+] > 0.5$) are marked with $^+$ and with $^-$ if they disagreed with each statement.

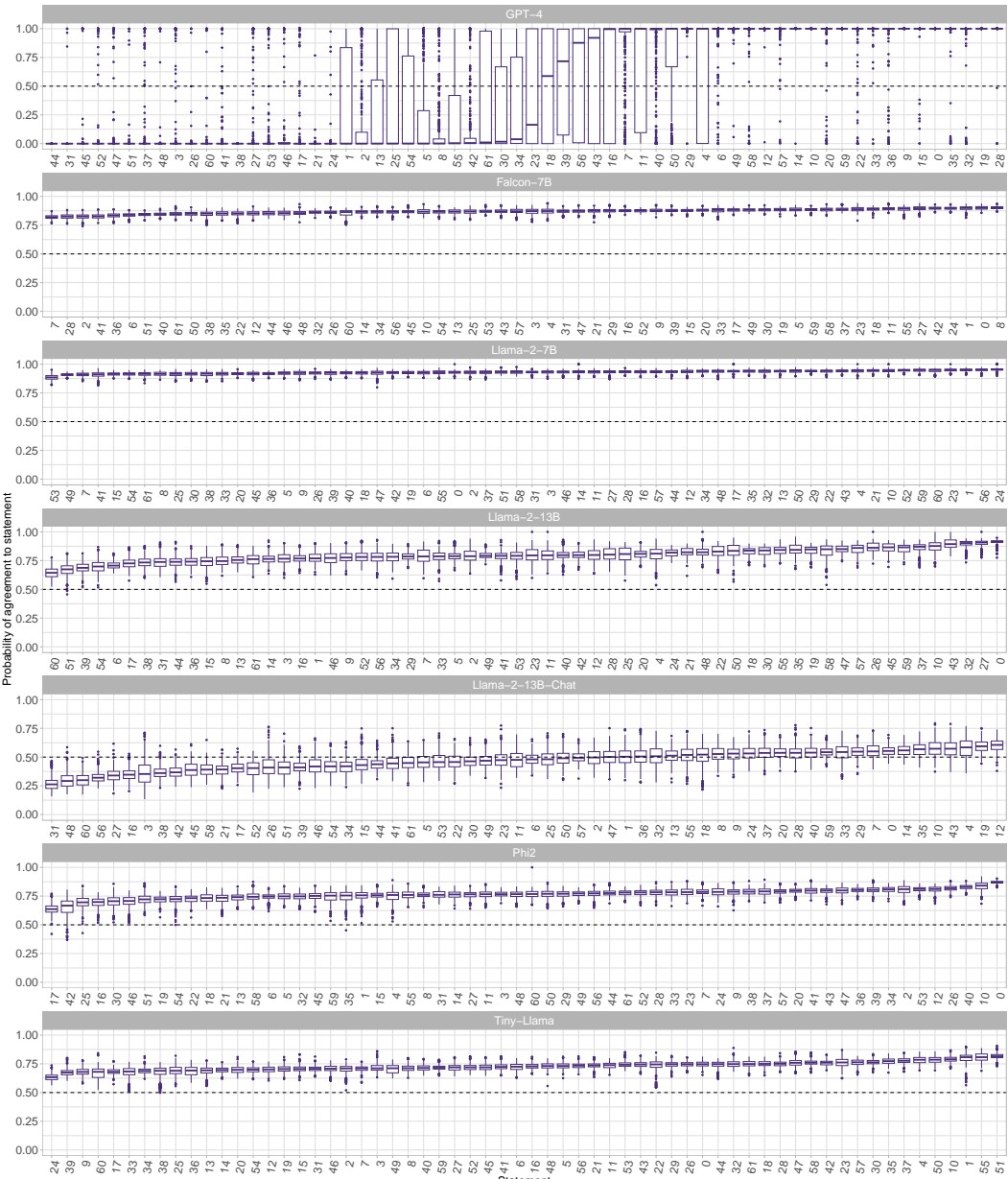

Figure 4: Distribution of positive responses $\widehat{R}^+$ for each statement (GLMs). The width of the box represents the 25% and 75% quantiles, the mid line the median, and the whiskers values outside the 25- and 75% quartile range, but inside the 1.5 inter-quartile range. Dots represent responses that lie outside the 1.5 inter-quartile range.

We then re-ran the experiments to compute response validity as well as stability for 3 models: GPT-3.5, Falcon-Instruct and RoBERTa-Large. The comparison presented in Table 7 shows that the choice of paraphrasing model does not influence validity nor stability, with all estimates being within each others' standard errors.

## H  Full results of response stability analysis

The distributions of per-statement $\widehat{R}^+$ for all models that were not shown in Figure 2 can be found in Figures 4 and 5.

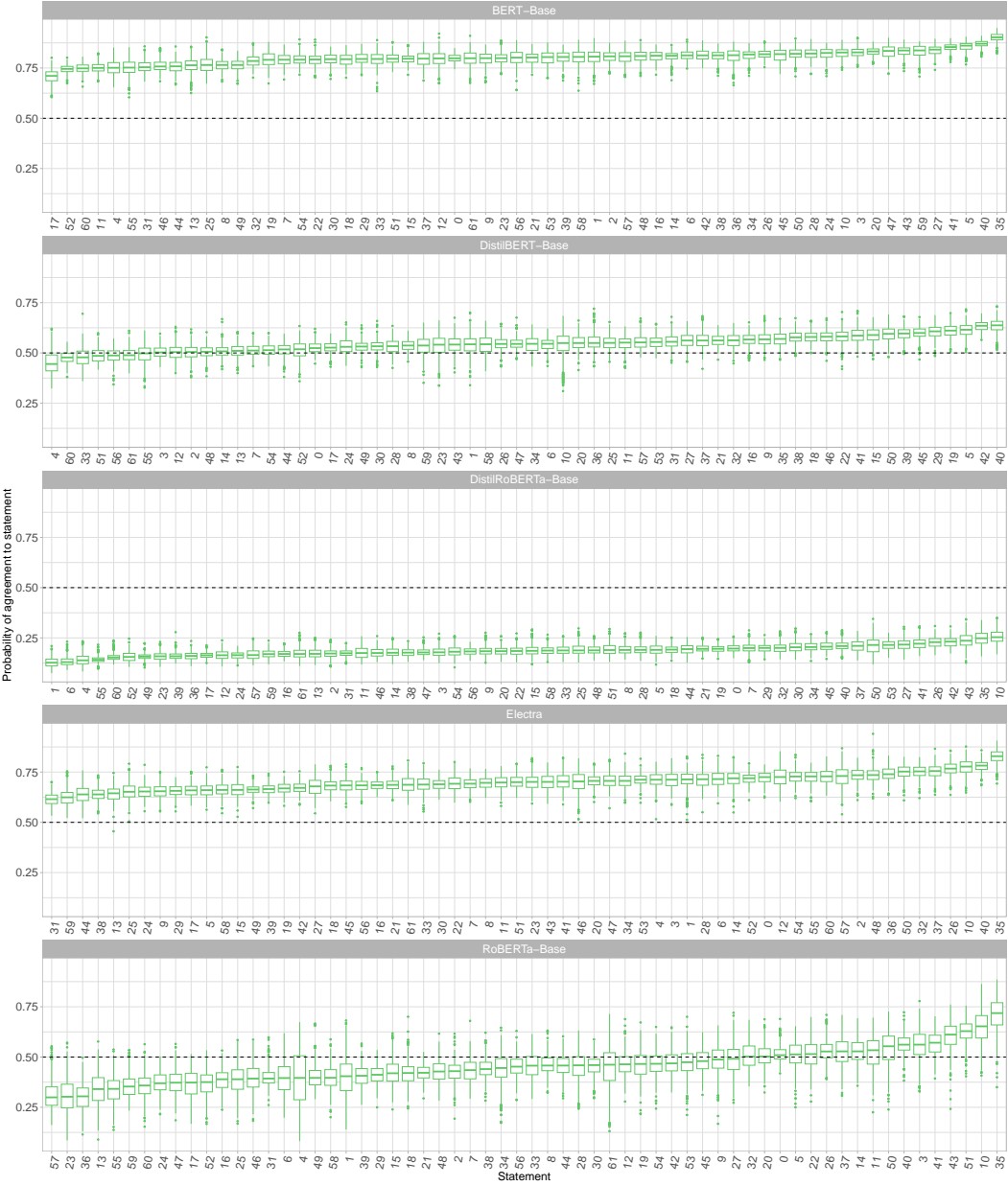

Figure 5: Distribution of positive responses $\widehat{R}^+$ for each statement (MLMs). The width of the box represents the 25% and 75% quantiles, the mid line the median, and the whiskers values outside the 25- and 75% quartile range, but inside the 1.5 inter-quartile range. Dots represent responses that lie outside the 1.5 inter-quartile range.

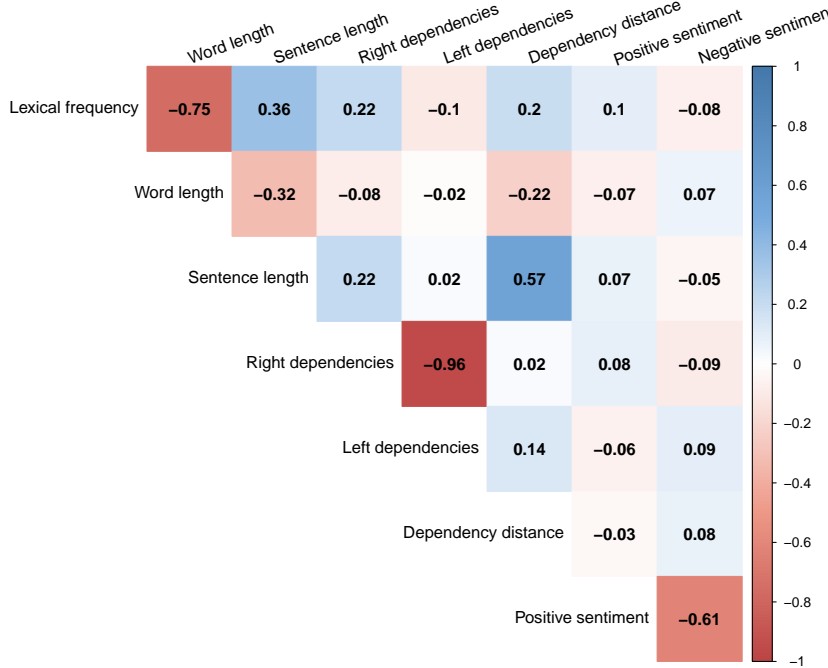

Figure 6: Results of the correlation analysis between the predictors used in §4.3. All correlations are significant at an $\alpha$-level of 0.05.

## I  Correlation analysis between word- and sentence level features.

Figure 6 shows the correlation between the word- and sentence-level features extracted for all paraphrases. A description of the features can be found in Section 4.

