# OpenReview forum: "Yes, no, maybe? Revisiting language models' response stability under paraphrasing for the assessment of political leaning"
_colmweb.org/COLM/2024/Conference — COLM_

### Official Review · Reviewer_Xcm7 · 2024-05-08

**Rating:** 7
**Confidence:** 4
**Ethics Flag:** 1

**Summary:**

This paper provides a large-scale analysis of LLMs consistency in answering paraphrased questions about their political leanings.
While the question of how stable and consistent LLMs’ responses to value questionnaires are has been (quite extensively) studied, this paper does it on a much larger scale, by covering 500 different paraphrases of each statement. Additionally, they also study linguistic factors which could bias models to respond in a certain way.
Overall this is a well-written paper, with extensive experiments and useful insights into LLM consistency.

**Reasons To Accept:**

- The scale on which the experiments, specifically the paraphrases, were performed: previous works performed these experiments on a much smaller scale, so it is great to see results over 500 different paraphrases per prompt
- The last experiment’s results, which was looking at possible features that could bias models, was interesting: The finding that models tend to agree with positive statements over negatively framed one.

**Reasons To Reject:**

- Despite citing works that criticized the forced-choice approach of the political compass setup, this work only evaluates on the Political Compass Test.
- As far as I could see there was no manual evaluation done on (a subset of) the 500 paraphrases, in order to check for quality, diversity and whether they still express the same semantic proposition.

---

> ### Author Rebuttal · Authors · 2024-05-31
>
> We thank reviewer Xcm7 for their review and appreciate the positive feedback.
>
> **Paraphrases**: It is correct that we did not check all 500x61 paraphrases manually.  However, we did randomly sample 10 paraphrases of each statement to verify if their propositions were consistent. For a (potential) revised version, we suggest increasing this sample size (e.g., 50) and reporting the number of “incorrect” paraphrases, if there are any.
>
> Regarding diversity, we somewhat take this into account by controlling for sentence length, average word length, and lexical frequency. To also control for syntactic diversity, we will include two additional predictors: the number of left dependents (i.e., how many dependents does a syntactic head control) and dependency length (i.e., distance of a dependent to its syntactic head) in the last experiment. Moreover, we will include descriptive statistics over these word-level features for the statements over paraphrases in the Appendix. That way, it will be possible to interpret the results in consideration of lexical and syntactic diversity.
>
> **Why the political compass test?**: Our approach to directly access the probability distribution over valid responses to a given statement required us to employ a test that uses forced choice, and the political compass was our test of choice as it makes our study comparable to previous work. We don’t disagree that forced choice is probably not a good way to probe abstract concepts such as political biases in LMs---this is one of the key take-aways of this work: the assessment methods developed for human subjects (often forced-choice) are not necessarily applicable to LMs. We will make this more clear in the (potential) revised version.

---

> > ### Comment · Reviewer_Xcm7 · 2024-06-04
> >
> > Thank you for your response! It would be great if you could include the promised revisions (i.e. number of incorrect paraphrases and descriptive statistics) in the camera ready. I think it's a good paper and am sticking to my current score.

---

### Official Review · Reviewer_Xq2x · 2024-05-09

**Rating:** 4
**Confidence:** 3
**Ethics Flag:** 1

**Summary:**

This paper analyzes shortcomings of using LMs for the assessment of political leaning. Specifically, the paper investigates model response stability using the Political Compass Test (62 statements with 4 options each). The authors focus on response variances on a per-item level under paraphrasing as well as which samples are particularly brittle across models and whether any linguistic and structural factors (e.g., word length, lexical frequencies) impact such findings. Using 500 paraphrases per model and 20 models in total (both MLMs and GLMs), the authors overall find that several models exhibit large ranges w.r.t. their response stability, and MLMs overall achieve a higher response validity as compared to GLMs. Moreover, the authors report that the sentiment of a statement consistently correlates with whether a model agrees to the statement, and both greater word and sentence counts are also correlated with agreement to the statement.

**Questions To Authors:**

Can you provide some details on specific model configurations and parameters that you used as part of the sampling? How would the results change when those configurations are changed?

**Reasons To Accept:**

* The paper sheds new light on the shortcomings related to the usage of LMs in the context of assessing political standpoints and personality traits. Since this topic has received increasing attention from the research community, work that critically assesses previously obtained findings is important and valuable.
* I agree that a robustness analysis on a larger scale (i.e., with a large amount of paraphrases of questions) can provide additional insights into the consistency of using LMs as respondents to such questionnaires.

**Reasons To Reject:**

* The paper generates paraphrases with an OpenAI model and evaluates OpenAI models on them, showing that they have the highest instruction validity rates. This experimental setup is inherently biased towards OpenAI models, and direct comparisons between OpenAI models and non-OpenAI models therefore become unfair. How would such results look like if a different model were used for paraphrasing?
* The paper does not discuss model configuration and sampling parameters (e.g., temperatures) and how these might potentially affect the obtained results. The probability distribution computed over potential outputs is directly affected by the temperature value and the results reported in Figure 2 would therefore directly shift as a result of changing that parameter. Additional work is needed to discuss and investigate the impact of such configuration changes on the obtained results.
* The paper does not discuss limitations of the proposed approach and obtained findings. Given that this work critically assesses the usage of LMs in the context of behavioral questionnaires, it would be important that the authors clearly address the scope and relevance of their work.
* [Style-related] Figure 2 is too small and difficult to read.

---

> ### Author Rebuttal · Authors · 2024-05-31
>
> We thank reviewer Xq2X for their critical assessment and the constructive feedback. We will address their concerns in the following.
>
> **Paraphrasing & Evaluation**: We agree that the choice of paraphrasing model will likely impact the results, due to differences in the quality or diversity of the paraphrases. Although the likelihoods of the statements per se might be higher for GPT-3.5, it is not evident why the next-token probabilities would be systematically biased towards the tokens mapped to “agreement” and “disagreement”. In the course of this project, we ran an analysis where we investigated whether models tend to agree with statements for which they assign higher likelihoods, and did not find evidence that this is the case. Nevertheless, we understand that this might also be a concern for readers, which is why we generated 500 additional paraphrases using Claude-3-haiku. The results show slightly higher instability *across all models*. We can discuss in the discussion period where to include these results.
>
> **Parameters**: We reported temperatures and sampling parameters for generating the paraphrases in Appendix D. For all other parameters, we used the default settings. We used this setting because it reliably provided diverse paraphrases that kept the semantic proposition.
> Other than that, we did not have to set any specific configurations since we were not *generating* text to elicit responses. We directly access the pre-trained models’ probability distribution conditioned on the statement and extract the probability for the tokens reflecting agreement or disagreement.
>
> **Limitations**: We are going to extend our discussion, including (but not limited to) the following points:
>
> * We worked with a fixed prompt template. Previous work has found that models indeed show instability in this respect as well.
> * We collapsed the response distribution from 4 responses to 2 (collapsing “strongly agree” and “agree”, for instance, which would now allow us to assess actual test results of the compass test. However, this was not the scope of this work, but we will include a discussion on how we concretely recommend to account for instability when obtaining actual test results (i.e., the location on the political axes).
> * We did not manually check each paraphrase to assess whether it reflects 100% the same semantic content as the original statement (but see response to reviewer  Xcm7).

---

> ### Author Response · Authors · 2024-06-05
> **Response to rebuttal**
>
> We are curious to hear reviewer Xq2X's response to our rebuttal. We dedicated a significant amount of time to address the issues concerned from their review. As written in our rebuttal, we believe that the two main points addressed in 'reasons to reject' were clarified sufficiently: As for point 1, we have explained why we don't think that our results are biased, and we nevertheless ran additional experiments to substantiate this. As for point 2, again, there were no model configuration and sampling parameters to discuss. Considering the clear deviation in reviewer Xq2X's score compared to the other reviewers, we believe it would be fair to hear a response to our rebuttal before the deadline.

---

> ### Comment · Reviewer_Xq2x · 2024-06-06
>
> I would like to thank the authors for their detailed response and the additional analysis. I appreciate that an additional set of paraphrases was generated with an independent model (i.e., one that is then not evaluated on the paraphrases). Based on the reasons provided, I would expect this to be the default experimental setup as it would exclude potential biases occurring from evaluating a model based on data that it generated. However, given that this would substantially change the reported results, this would exceed the scope of a revision after reviews.
>
> Re. the parameters: I appreciate the clarification and agree that in this case no additional information would be required (misunderstanding on my side).
>
> I also appreciate the additional limitations provided.
>
> Based on the above I increased my score, but still believe that the paper would benefit from an additional round of revision.

---

### Official Review · Reviewer_WiWK · 2024-05-10

**Rating:** 7
**Confidence:** 4
**Ethics Flag:** 1

**Summary:**

This paper takes on one small piece of the now familiar setup, in which psychometric instruments or political ideology instruments are given to a LLM to measure something about them (though specifically what we are supposed to infer from such measurements is often left rather vague, as is the case here).

Although there are many such variations, this paper uses a fairly familiar one, which is to set up a prompt with one space for a statement, and another for the model response (e.g,. "Do you agree or disagree with the following statement: <statement>. I <response> with this statement." for a MLM). They also use a common measurement strategy, which is to sum the probabilities associated with, roughly, affirmative or negative responses.

 Although there are many things which might be varied in such a setup, here the authors focus on the phrasing of the statement itself (such as "There are those who are destined to experience chronic misfortune"). While others have tried various systematic modifications of statements, here the authors simply collect a large number of paraphrases of each statement, and check to see how much probabilities vary across paraphrases.

Ignoring some of the nuance, overall the authors find that models are relatively stable, but with some showing a large range between the min and max probabilities produced by different paraphrases of the same statement. Thus, as with other related papers, the authors provide additional evidence that this overall approach is not necessarily a valid way to infer something like ideology or personality for LLMs.

**Reasons To Accept:**

This paper is clear and well written. Although there are strong prior reasons why most people would naturally be skeptical of the validity of trying to assess LLM "personality" using psychometric instruments, it is a popular enough idea, that papers such as this are important for pushing back against it.

Although this paper provides less insight into the types of modifications that can lead to instability, its strength lies in capturing some sense of the variation we might expect across reasonable paraphrases of the statements (conditional on one particular prompt and decoding setup, each of which is also likely to contribute to additional variation, possibly in a way that interacts with what is measured here).

The authors test a large number of models, and include a few examples, although many more would be welcome.

Overall this is not the most exciting paper on its own, but it adds to the weight of evidence that can be dervied from the many similar papers.

**Reasons To Reject:**

For experimental setups, one necessarily has to fix something (e.g, prompt, decoding strategy, etc.) to have any level of tractability. And yet the fact that these things are fixed here means that we don't really know how well these results will generalize to other choices that could have been made. Compared to something more like a physical system, it seems less clear that we can count on interactions between such pieces of the experimental setup to not matter.

The authors do attempt to provide some sort of analysis of their results, using linear models with factors such as sentiment and paraphrase length, to see if these are predictive of the results, but these are less compelling than the rest of the paper, in part because we would expect such things to be highly contextual. (They find, for example, that, in their setup, models are more likely to agree with statements that have positive sentiment).

Other issues are mostly minor. In particular, many more examples of prompts and variation (ideally a large random selection) would be extremely welcome. Some of the figures (e.g., Figure 2), involve a lot of text that is far too small to be legible and/or a lack of context (e.g., numbered items without a legend or key), undermining their utility.

---

> ### Author Rebuttal · Authors · 2024-05-31
>
> We thank reviewer WiWK for their careful review and appreciate their positive feedback.
>
> **Generalization to other experimental settings**: The advantage of our proposed method where we directly access the model’s probability distribution instead of generating the responses is that there is no need to fix generation parameters such as decoding strategy and its hyper-parameters. While it is correct that we do not know whether the analyzed linguistic features will show the same effects when varying the prompt, we contend that in particular, the results for sentiment were consistent across all tested models (MLM & GLM), even though they may all require specific prompt templates.
> We agree that many more examples of prompt variations would be an interesting additional dimension to consider. For a potential revision, we started setting up an experiment where we will verify the stability of the results for 9 prompt variations (3 formats: changing new lines/spaces, etc. x 3 content-level variations: different questions).
>
> **Minor issues**: Regarding the figures, we will increase the axis labels of the plots. We were unsure about the copy-right of the political compass statements, which is why we hesitated to include them. We are in the process of clarifying the copyright situation and, if possible, add a table of all original statements in the Appendix and will also release all paraphrases in the final repository.

---

> > ### Comment · Reviewer_WiWK · 2024-06-06
> > **acknowledgement**
> >
> > Thank you for your response. I recognize that there is some disagreement among reviewers as to this paper, but I will keep my score as it is, which I think is a reasonable assessment. I will look forward to the revised paper, and I hope you are able to resolve the copyright question.

---

### Official Review · Reviewer_bsVC · 2024-05-14

**Rating:** 5
**Confidence:** 4
**Ethics Flag:** 1

**Summary:**

The paper presents a large-scale empirical study on stability (sensibility and consistency) around paraphrasing abilities in LLMs in relation to political leanings. Unlike previous studies of similar inquiry, this work conducted deeper statistical analysis of data, thanks to the larger sample size and scope. The authors conclude that tests designed for bias measurement among humans (e.g., Political Compass Test) are likely ineffective in measuring bias in LLM due to the instabilities in their responses.

Political and social bias in generative capabilities in LLM is an important and timely issue, both as a practical concern as well as a theoretical exercise. Overall the paper is written well and the points come across very easily. The proposed methodology is sensible and executed well. I however wonder how novel the techniques and applications are at this time, and how they are generally usable/useful in other similar inquiries. Even before Feng et al's seminal 2023 papers, political bias in LLM has been a concern among the practitioners. This paper contributes to the body or research, though I wonder whether there are enough generalizable technical innoventions and/or interesting or surprising findings. There are several promising analyses (e.g., stability in relation to text features) but I think they need to be developed more. Another point is a practical application of the work; I think the paper would be much stronger if the authors elaborated on how the presented methods (or the findings) are useful in real world tasks.

**Questions To Authors:**

NA

**Reasons To Accept:**

* clear writing
* some interesting finding
* solid analysis, some interesting follow ups.

**Reasons To Reject:**

* lack of novelty in idea
* lack of technical invention
* lack of practical application or next research direction
* not enough new discovery

---

> ### Author Rebuttal · Authors · 2024-05-31
>
> We thank reviewer bsVC for their critical assessment. We understand their concerns and would like to address them in the following.
>
> **Novelty and usefulness**: Although we acknowledge that our technique is not novel per se, we’d like to emphasize that unlike other approaches, our method of eliciting a model response does not rely on directly sampling model responses, as we directly access the distribution conditioned on the context (i.e., the statement). While the paraphrasing of the statements might not be a technical innovation, to the best of our knowledge, we are the first to assess its utility at large-scale, and we believe it is a simple but effective, computational less costly, and powerful method to obtain statistically more powerful results in the context of model probing that is applicable in similar cases as well.
>
> **Utility in real-world tasks**: We acknowledge that we haven’t elaborated about the utility in real-world tasks. We believe that the usefulness of our work might not lie in a direct application, but rather illustrates the difficulty of assessing higher-level / abstract concepts such as political biases in LMs. Although currently, only a few models are extensively used by a wide public, in the future, many more might contend for popularity. We believe that it is important to have tools and techniques that reliably elicit properties and characteristics of LMs that public or private companies would like to be informed about. As it is for now, our work shows that the assessment methods developed for human subjects are not necessarily applicable to LMs, i.e., and secondly, that it is important to consider stability on the item-level in addition to overall model-stability. Researchers will continue using tests developed for human participants, and we think that it is important to present evidence that such studies only make sense if the results are evaluated by taking into account model uncertainty or using new techniques altogether. We will discuss this point and add this as a concrete recommendation for future work in our revised version.
>
> **Promising analyses but need to be developed more**: As elaborated in the response to reviewer Xcm7, we aim to additionally include two additional predictors to explore the impact of syntactic features on the response distribution: number of left dependents (i.e., how many dependents does a syntactic head control) and dependency length (i.e., distance of a dependent to its syntactic head).

---

### Author Response · Authors · 2024-06-03
**Discussion period**

We again thank all the reviewers for their feedback and look forward to discussing the points raised in the reviews and rebuttals.

---

### Author Response · Authors · 2024-06-07
**Thanks to reviewers**

We thank the reviewers for engaging in the discussion period and responding to our rebuttals, we appreciate it.

---

### Decision · Program_Chairs · 2024-07-10

**Decision:**

Accept

**Comment:**

This paper reports a large-scale study on stability in light of paraphrasing with respect to political leanings, concluding that due to instabilities tests that are used to measure bias in humans may not be effective for LLMs.

Reviewers appreciate that the paper addresses a worthwhile topic, provides some interesting insights, and performs these analyses at larger scale than prior work.

However, reviewers also express some lukewarm feelings about the overall impact and excitement level for this paper -- the question asked here has been investigated often in prior work, and the main novelty here seems to be the larger scale at which the investigation is conducted. There are also some questions about to what extent the findings generalize beyond the specific setup (e.g., prompt) used here, as well as lack of clarity about practical applications. Finally, a concern is also raised about overlap between OpenAI models used for generating paraphrases and the models being evaluated in the analysis -- and though the authors report a follow-up analysis in response to this concern, I agree that it would be nice for this to be integrated more centrally in the paper.

All in all, this seems like a solid and safely publishable work, but it could probably be strengthened in another round of revisions, integrating the analysis without the paraphrase model among the evaluated models, making stronger arguments for practical utility, and ideally providing further evidence for generalizability and novel impact.